# Trade informativeness and liquidity in Bitcoin markets

J. Christopher Westland [ID]*

University of Illinois Chicago, Chicago, Illinois, United States of America

* westland@uic.edu

## Abstract

Liquid markets are driven by information asymmetries and the injection of new information in trades into market prices. Where market matching uses an electronic limit order book (LOB), limit orders traders may make suboptimal price and trade decisions based on new but incomplete information arriving with market orders. This paper measures the information asymmetries in Bitcoin trading limit order books on the Kraken platform, and compares these to prior studies on equities LOB markets. In limit order book markets, traders have the option of waiting to supply liquidity through limit orders, or immediately demanding liquidity through market orders or aggressively priced limit orders. In my multivariate analysis, I control for volatility, trading volume, trading intensity and order imbalance to isolate the effect of trade informativeness on book liquidity. The current research offers the first empirical study of Glosten (1994) to yield a positive, and credibly large transaction cost parameter. Trade and LOB datasets in this study were several orders of magnitude larger than any of the prior studies. Given the poor small sample properties of GMM, it is likely that this substantial increase in size of datasets is essential for validating the model. The research strongly supports Glosten's seminal theoretical model of limit order book markets, showing that these are valid models of Bitcoin markets. This research empirically tested and confirmed trade informativeness as a prime driver of market liquidity in the Bitcoin market.

## 1. Introduction

Liquidity is a measure of a market's ability to address the demands of impatient traders. Liquidity demanders are more likely to be privately-informed, through research or inside knowledge of the market, than are passive liquidity suppliers, who may be more concerned with price stability and predictability [1]. [2] show that where there is a higher chance of informed trading, we can expect higher returns in the form of a volatility risk premium. [3] demonstrated why markets need uninformed and informed traders—the volatility in prices and volume brought by uninformed liquidity suppliers makes continuous profit possible for informed traders. [4] uses the metaphor of "sharks" (informed players) and fishes" (uninformed players) in poker to illustrate how these information asymmetries drive financial markets. A liquid market requires an unending supply of "fish" if the "sharks" are to make a profit. Fish are willing to make price concessions to "sharks" as a way of lowering their risk. They

**Data Availability Statement:** The data are held or will be held in a public repository, on Kaggle at James Christopher Westland, "Bitcoin Market Trade Informativeness and Liquidity." Kaggle, 2021, doi: 10.34740/KAGGLE/DSV/2190153.

**Funding:** The author received no specific funding for this work.

**Competing interests:** The author has declared that no competing interests exist.

tend to panic and fold too early, especially when they have money committed, leading to a steady flow of revenue into the sharks' pockets.

More than half of asset markets, including most cryptocurrency markets, now use an electronic limit order book. This was not the norm when [5] presented his seminal model of an electronic limit order book market, but since that time, the major asset markets around the world have implemented electronic limit order book systems. Cryoptocurrency markets invariably use electronic limit order books with relatively low transaction costs and high volumes. Cryptocurrency traders have a rich collection of order choices including limit, stop limit, market, and various derivatives. Each of these supplies of demands liquidity in specific ways.

Many types of information fuel information asymmetries in Bitcoin markets: e.g., demand to convert other assets to Bitcoin; changes in the total supply of Bitcoins through mining and recirculation of 'dark pools'; uncertainties in reserve currencies such as the dollar, where, like gold, Bitcoins may be seen as a 'safe haven' from macroeconomic uncertainties, and so forth. Bitcoin's supply is algorithmically capped at 21 million, out of which around 19 million Bitcoin have already been mined of which Satoshi Nakamoto owns around 1.1 million [6] and another 3.7 million Bitcoins have not been used in the past 5 years [7] all of which contribute to price volatility. In addition, around 2700 Bitcoin have been sent to 'burn addresses'—vanity addresses with no known private key—and are likely out of circulation, along with several thousand inaccessible Bitcoin belonging to deceased owners who left no records [6]. There has been a rapid growth in derivatives, and now about one-third of the volume of cryptocurrency trading has moved into derivatives markets with their much greater volatility. Regulation of cryptocurrencies is rapidly evolving, and generally seeks market transparency and taxation, features that cryptocurrencies may systematically try to thwart [8].

The current paper studies the impact of private and public information on cryptocurrency prices and trading, using [5] model of electronic limit order book (LOB) markets. Section 2 reviews the prior literature in cryptocurrency and LOB markets. Section 3 describes the datasets used in the empirical studies in this paper. Section 4 presents a structural model based on [5] that is used for estimation with these datasets. Section 5 reports the results of model fitting, and Section 6 discusses the implications of this research.

## 2. Prior literature

Limit order book econometric models may be either static or dynamic. Static models are supply-demand equilibrium models, where private information is injected into a liquidity-providing market only on the demand side. The limit order book determines the supply inventory, and demanders arrive randomly to appropriate a portion of the supply through aggressively priced market orders. [5] provided the seminal electronic order book static model where risk-neutral limit order posters compete for supply, and the market clears where there is no excess profit to be gained. A follow-up study by [1] limited the market participation of strategic suppliers, a model that converges to the [5] results asymptotically. Parallels appear in [9, 10] who consider NYSE-type markets with specialist functions whose role is price stabilization and injecting more liquidity. [11] found that compared with such specialist-enabled markets, pure electronic limit order book markets improve the competitive equilibrium obtained. [12] concluded that in electronic limit order book markets, market orders are a primary point of injection of private information. [13] found, in the hybrid NYSE market, that specialists and floor brokers do indeed trade on superior private information.

In contrast, dynamic models start with a queue of undifferentiated traders who want to use the market. Impatient traders are willing to submit market orders for immediate execution for

a risk-premium equal to the bid-ask spread. Limit order traders may wait forever for a trade, while market order traders experience near-zero delay, while injecting new information into the market every time a market trade occurs. This injection of risk information into the market is called "picking-off risk" and causes limit orders to execute more often with higher price variance in the market.

[14] tested the [5] model for stocks traded on the Stockholm Stock Exchange (SSE). The SSE is a relatively simple and thinly traded market, thus the inherently asymptotic results of the Glosten model failed to obtain. The massively larger volumes in cryptocurrency markets should converge to Glosten's outcomes, a hypothesis that the current research tests. I believe that cryptocurrency data is likely to fit Glosten's models better, for two reasons:

1. both limit and market orders have the opportunity to make markets, injecting private information into their markets, because cryptocurrency fees and commissions are rapidly approaching zero, and are orders of magnitude smaller than fees charged in asset markets in the 1990s.

2. limit order activity is rapidly increasing because of radically lower fees and technological development. In earlier research by [14] the ratio of limit to market orders was 1.7; asset markets typically have ratios of 5 to 10; in the current research, the Kraken-Bitcoin ratio of limit to market orders is around 20

## 3. Dataset: Bitcoin trading data from the Kraken platform

Kraken is the second largest cryptocurrency exchange in the US by capitalization, and supports both limit and market orders for Bitcoin, as well as short sales and derivatives. Kraken is considered to be more technically sophisticated than CoinBase, and attracts more informed traders. Globally the top 4 exchanges (based on daily volume) are Binance ($ 5.46B), Huobi ($ 3.40 B) Coinbase ($ 0.35B) and Kraken ($ 0.21) from coinmarketcap. Coinbase is more beginner-friendly than Kraken while Kraken has a wider selection of cryptocurrencies.

I analyzed 92,804,000 trades and 18,943,200 limit orders from early August, 2020, organized into six tranches for Bitcoin on the Kraken exchange to empirically assess liquidity. These dates were predicted to be "information rich" for two reasons: (1) global investors and cryptocurrency traders were cashing in on some of their profits, as the cryptocurrency market is washed with cheap money coming from stimulus packages, and Bitcoin prices reached historical highs; and (2) at 8:28AM on Aug 6 a bitcoin 'whale' transferred 92857 BTC ($\approx$\$1.1 *billion*) between two wallets, setting off several days of speculation [15].

I acquired these datasets directly from Kraken's native RESTful APIs using custom code. Kraken is throttled to protect against DDOS attacks, and the code dealt with that, as well as the nanosecond resolution of trade times, which is too small a resolution for standard software arithmetic to handle. The data consists of a baseline 32 hours of data, and five datasets of 2-5 hours of trading. The large number of limit orders enables the assessment of information content of market orders. On average, the order book accumulated around 20 limit orders between each market order, which is substantially in excess of similar figures in traditional asset markets, which typically accumulate 5 to 10 limit orders per market order, reflecting lower charges in general for limit orders in cryptocurrency markets.

Fig 1 shows that market orders have more aggressive price moves than limit orders, supporting the idea that these trades are confidently made on new information available to the trader.

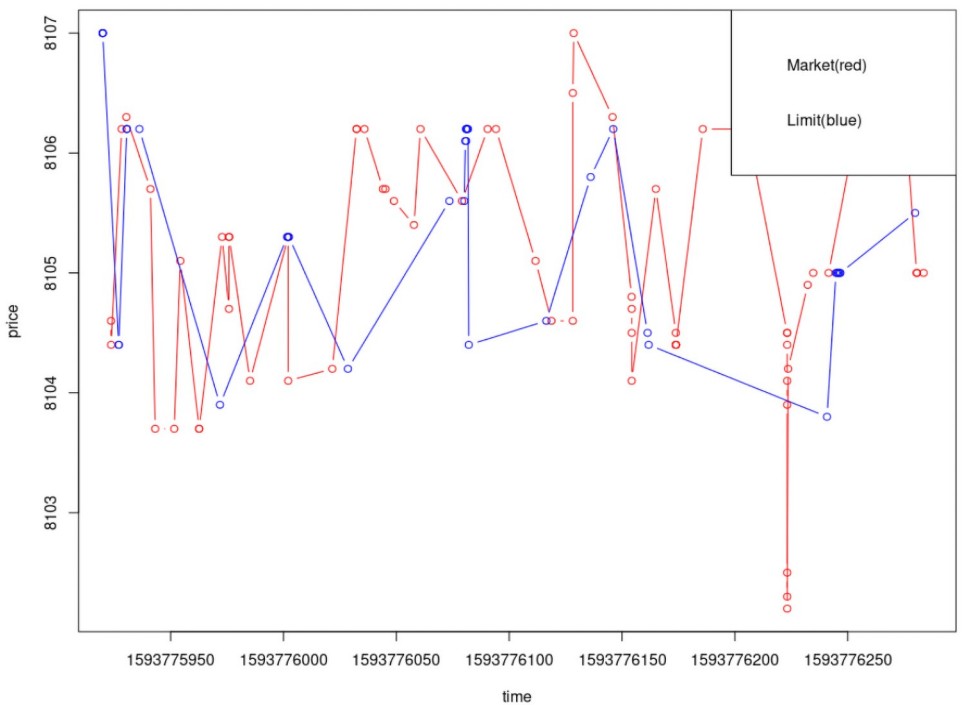

**Fig 1. Market orders display significantly more aggressive price moves than limit orders.**

Interestingly, limit orders seem to show more volume swings than market orders (Fig 2), suggesting two things: (1) market orders are concerned that they will move the price (unfavorably) too much, and thus tend to trade in small blocks, and (2) limit orders represent a portfolio optimizer's "wish list," and where the "wish" is executed, they want to buy or sell as much at that price as they can or have in inventory. Algorithmic trading has grown in importance since the early 1990s with an explosion of electronic trading platforms after the 1987 market crash.

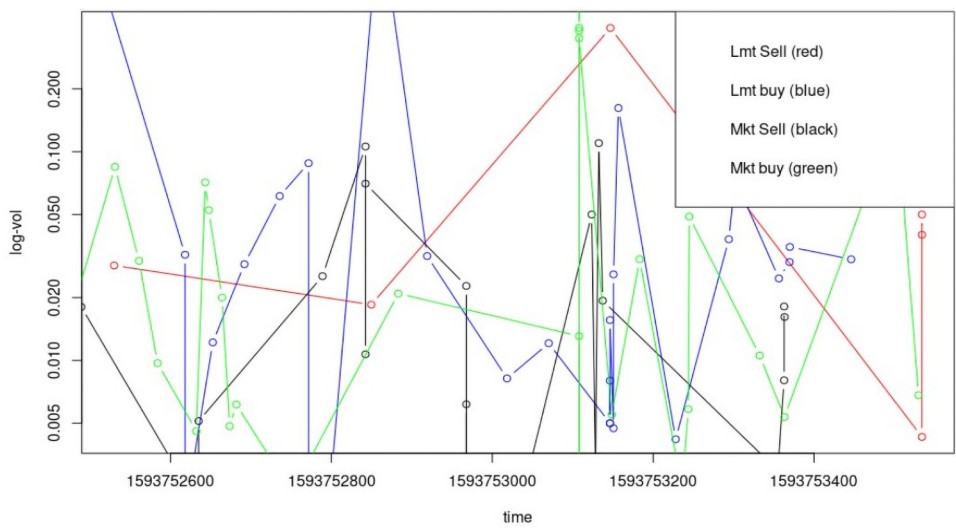

**Fig 2. Bitcoin trade dynamics (volumes of 200 trades).**

## 4. Glosten's structural model

This research followed [14] of the [5] structural model. The Glosten model interestingly implies both forward and reverse Granger causality. For example in the case of an upcoming press release, press-release induced order flow may cause an immediate quote update and portfolio rebalancing [16] (Table 1).

[17] introduced the two-step generalized method of moments (GMM) to applied economics and finance, where it provides a generic method for estimating finite-dimensional parameters in semi-parametric models. GMM starts by positing a centralized moment condition, a system of $q \times 1$ potentially nonlinear equations $E[g(\theta_0, x_i)] = 0$ used to estimate parameters $\theta_0 \in \Re^p$. Boundary conditions may additionally be specified to insure a unique solution.

Fig 3 schematically describes the operation of market orders in a Glosten market. Trading events are assumed to arrive randomly, and in the period between market orders, limit order traders post to the LOB attempting to adjust their portfolios; illustrated in the following timeline:

where:

- $X_t$ is the market order size,

- $v_t$ is the true value of the cryptocurrency after a market order arrives, and

- $Z_t$ represents the current state of the LOB.

Market orders of size $X_t$ arrive at $t$, buys (positive) and sells (negative) have the same probability of occurring, and $X_t \perp\!\!\!\perp X_s$ for $s \neq t$. Order size is monotonic increasing on waiting

**Table 1. Modeling parameters used in the Glosten structural model.**

| Parameter | Description |
|---|---|
| $p_{i,t}$ | price of the $i^{th}$ best order (asks $i > 0$; bids $i < 0$) at time $t$ |
| $q_{i,t}$ | quantity of the $i^{th}$ best order (asks $i > 0$; bids $i < 0$) at time $t$ |
| $X_t$ | the market order size (positive or negative) |
| $Z_t$ | the state of the order book |
| $v_t$ | the true value of the asset (Bitcoin) after a market order $X_t$ arrives, and |
| $v_t = c + v_{t-1} + \alpha X_t + \eta_t$ | asset value update formula |
| $\alpha$ | key modeling parameter which measures the average information content of arriving market orders |
| $c$ | a consumption parameter that is set by the underlying market |
| $\eta_t$ | effect of information that arrives between trade times $t - 1$ and $t$ |
| $\lambda$ | expected absolute value of the limit order volume |
| $\gamma$ | the fixed order-processing cost of incoming market orders |

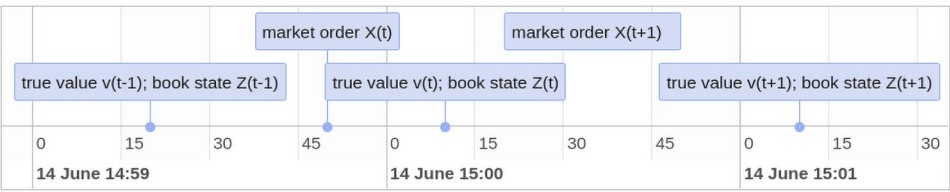

**Fig 3. Trading event random arrival.**

time, since traders have the option of splitting or consolidating orders as their private values change, thus a two-sided exponential distribution for waiting times is the most appropriate assumption:

$$f(X_t) = \frac{1}{2\lambda} e^{\frac{|X_t|}{\lambda}}$$

where $\lambda > 0$ is mean order size. The central parameter of the research is $\alpha$ which captures information about the underlying 'true' asset value from an arriving market order, i.e., how much of that information is impacted into the trading mechanism with the arriving market order. Thus, true value $v_t$ after the market order arrives at $t$ is:

$$v_t = E[v_t|v_{t-1}, X_t] + \eta_t = c + v_{t-1} + \alpha X_t + \eta_t \tag{1}$$

where:

- $\eta_t$ reflects information that arrives between trade times $t - 1$ and $t$.

The model assumes fixed limit order processing cost $\gamma$ and the various prices of a market order of volume $X_t$ elicits a response of limit order postings to the LOB until breakeven. For example, let $p_{1,t} \geq v_t$ be the lowest price at which it is advantageous to supply a limit sell order. Limit orders will be posted to the LOB up to and at this price. The expected profit on the $q_{1,t}$–th share at price level $p_{1,t}$ is given by:

$$E[(p_{1,t} - E[v_{t+1}|X_{t+1}] - \gamma) \times I_{[X_{t+1}>q_{1,t}]}] \tag{2}$$

where $E[(p_{1,t} - E[v_{t+1}|X_{t+1}] - \gamma)$ expected markup from true value, and conditional on the next market order $X_{t+1}$

- ($I_{[Xt+1 > q_{1,t}]}$ is $1$ if $X_{t+1} > q_{1,t}$ in which case the limit order executes)

Orders arrive at the market up to the point at which the 'true' value is reflected in the last limit order, or a trade clears:

- this process generates an equilibrium depth of $q_{1,t}$ at the best ask price $p_{1,t}$

- the next order arrives at one tick above $p_{1,t}$ generating potential revenue on execution is one tick higher than on $p_{1,t}$.

Fig 4 provides a plot of how these decisions happen in ~100 trades in my dataset from Kraken's July 6th 2020 Bitcoin trading. Notice the behavior of limit buy or sell trades (blue and red lines) after the price point set by a market buy or sell trade (green and black lines). Fig 5 provides a broader snapshot of the limit order book (top) and the actual execution of market and limit orders in the same period. Fig 6 zooms in on the order book's best four orders on either side of the market price. Taken together Figs 4 through 6 provide a detailed snapshot, using empirical data, of the trading processes in Glosten's model.

Thus the LOB state at any point in time $t$ is:

- bid $p_{-k,\,t}$ and ask $p_{+k,t}$ prices for $k = 1, 2, 3, \ldots$ and

- depths ($q_{-k,\,t}$ and $q_{+k,t}$) for $k = 1, 2, 3, \ldots$

The equilibrium equations show that the information injected into the market during trades is measured by $\alpha$ which is a key determinant of liquidity. The following recursions define the depths (and thus state of the LOB) on both sides of the LOB, with $\alpha$ as a key

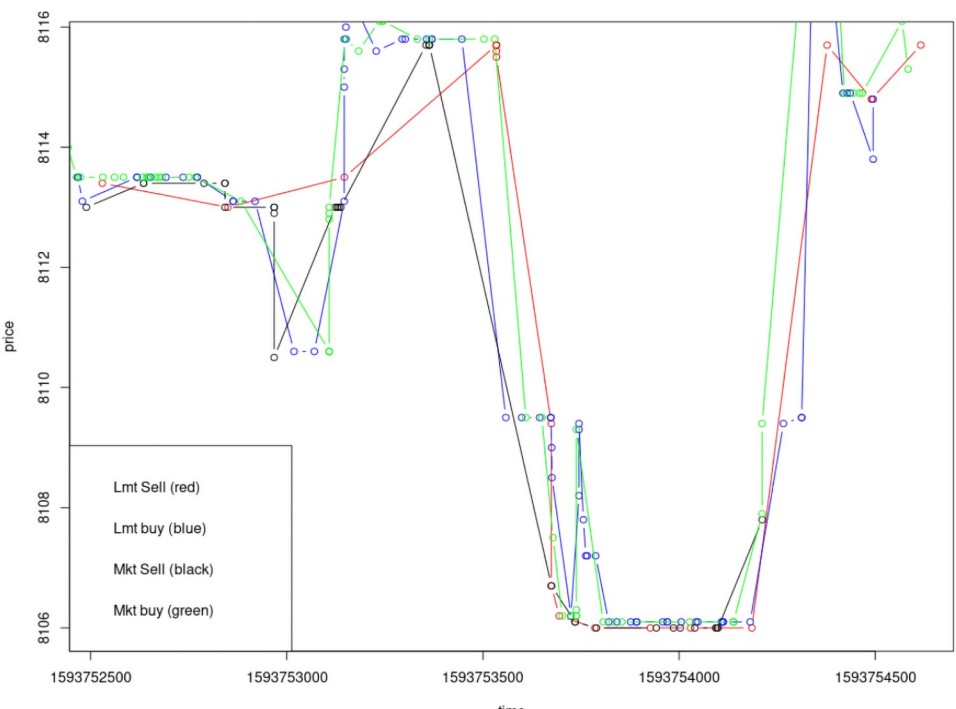

**Fig 4. Trading behavior of limit orders placed following a market order execution.**

determinant of book liquidity:

$$q_{-k,t} = \frac{v_t - p_{-k,t} - \gamma}{\alpha} - \sum_{i=-1}^{-k-1} q_{i,t} - \lambda \qquad k = 1, 2, \ldots \quad (bid \quad side) \tag{3}$$

$$q_{+k,t} = \frac{p_{+k,t} - v_t - \gamma}{\alpha} - \sum_{i=+1}^{+k-1} q_{i,t} - \lambda \qquad k = 1, 2, \ldots \quad (ask \quad side) \tag{4}$$

## 4.1 Glosten Model Moment conditions for GMM estimation

I followed Sandås (2001) model using three sets of moment conditions: two of these are based on Eq (4) where limit orders are posted to the LOB until equilibrium price, and then we take a snapshot at time $t$ just before the arrival of the next market order $X_{t+1}$. The third condition sets expected market order size equal to some fixed $\lambda$.

The **break-even moment conditions** pulls information from the LOB and removes the fundamental true value by adding the equilibrium depth associated with the $k^{th}$ price at the bid side of the book to the same equilibrium equation at the ask side of the book. We assume that these equations hold up to an error term:

$$E\left(p_{+k,t} - p_{-k,t} - 2\gamma - \alpha\left(\sum_{i=+1}^{+k} q_{i,t} + \sum_{i=-1}^{-k} q_{i,t} + 2\lambda\right)\right) = 0 \qquad k = 1, 2, \ldots \tag{5}$$

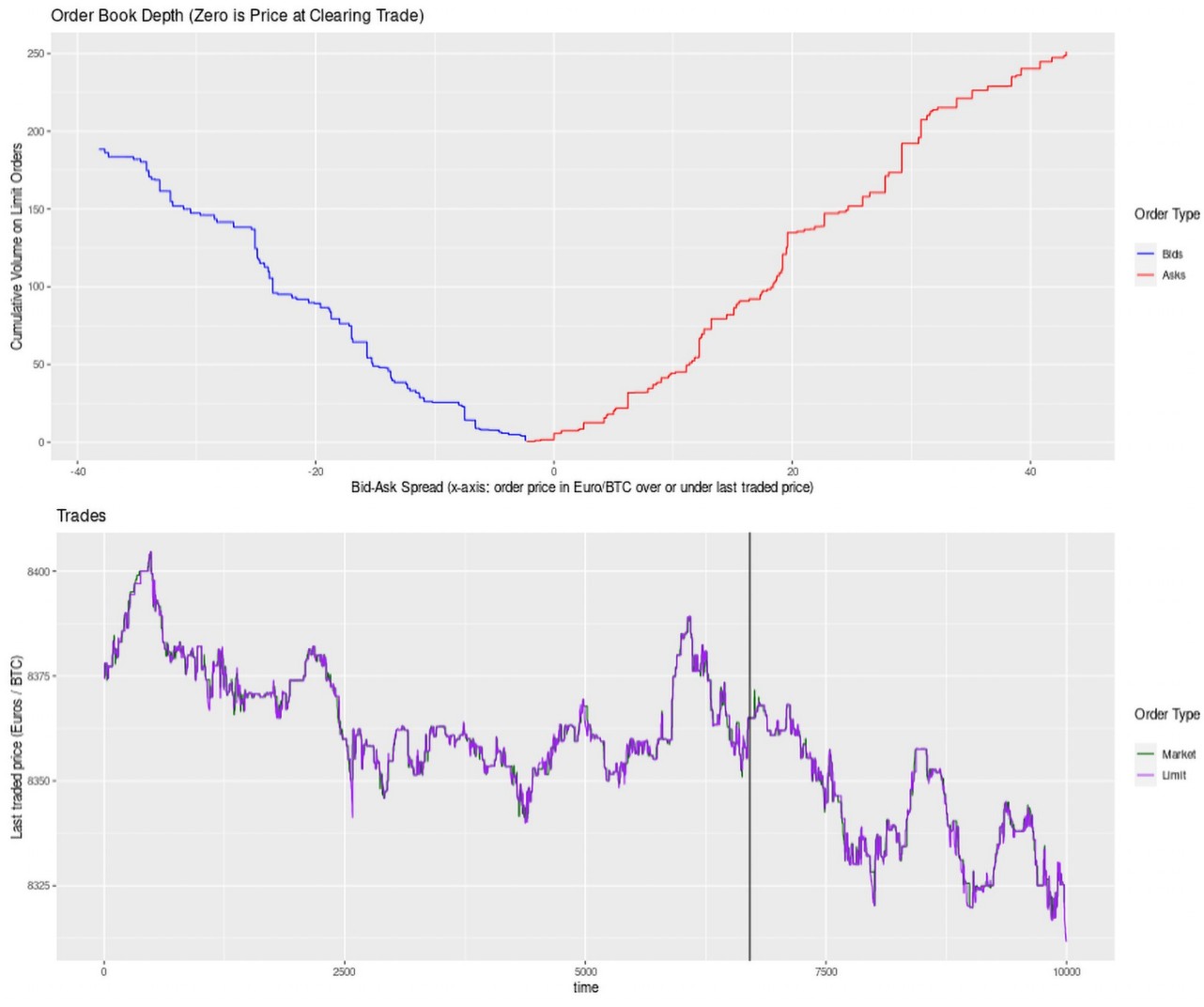

**Fig 5. Depth of full order book and market price.**

The **_updating restriction moment conditions_** subtracts $p_{\pm k,\,t+1}$ from $p_{\pm k,\,t}$ removing the 'true' asset value $v_t$ giving:

$$E\left(\Delta p_{+k,t} - \alpha\left(\sum_{i=+1}^{+k} q_{i,t+1} - \sum_{i=+1}^{+k} q_{i,t} - c - \alpha X_t\right)\right) = 0 \qquad k = 1, 2, \ldots \quad (ask \quad side) \quad (6)$$

$$E\left(\Delta p_{-k,t} - \alpha\left(\sum_{i=-1}^{-k} q_{i,t+1} - \sum_{i=-1}^{-k} q_{i,t} - c - \alpha X_t\right)\right) = 0 \qquad k = 1, 2, \ldots \quad (bid \quad side) \quad (7)$$

where $\Delta p_{k,t} = p_{k,t} - p_{k,t-1}$

**_Market order size conditions_** set $\lambda$ equal to the expected size of market orders:

$$E(|X_t| - \lambda) = 0 \tag{8}$$

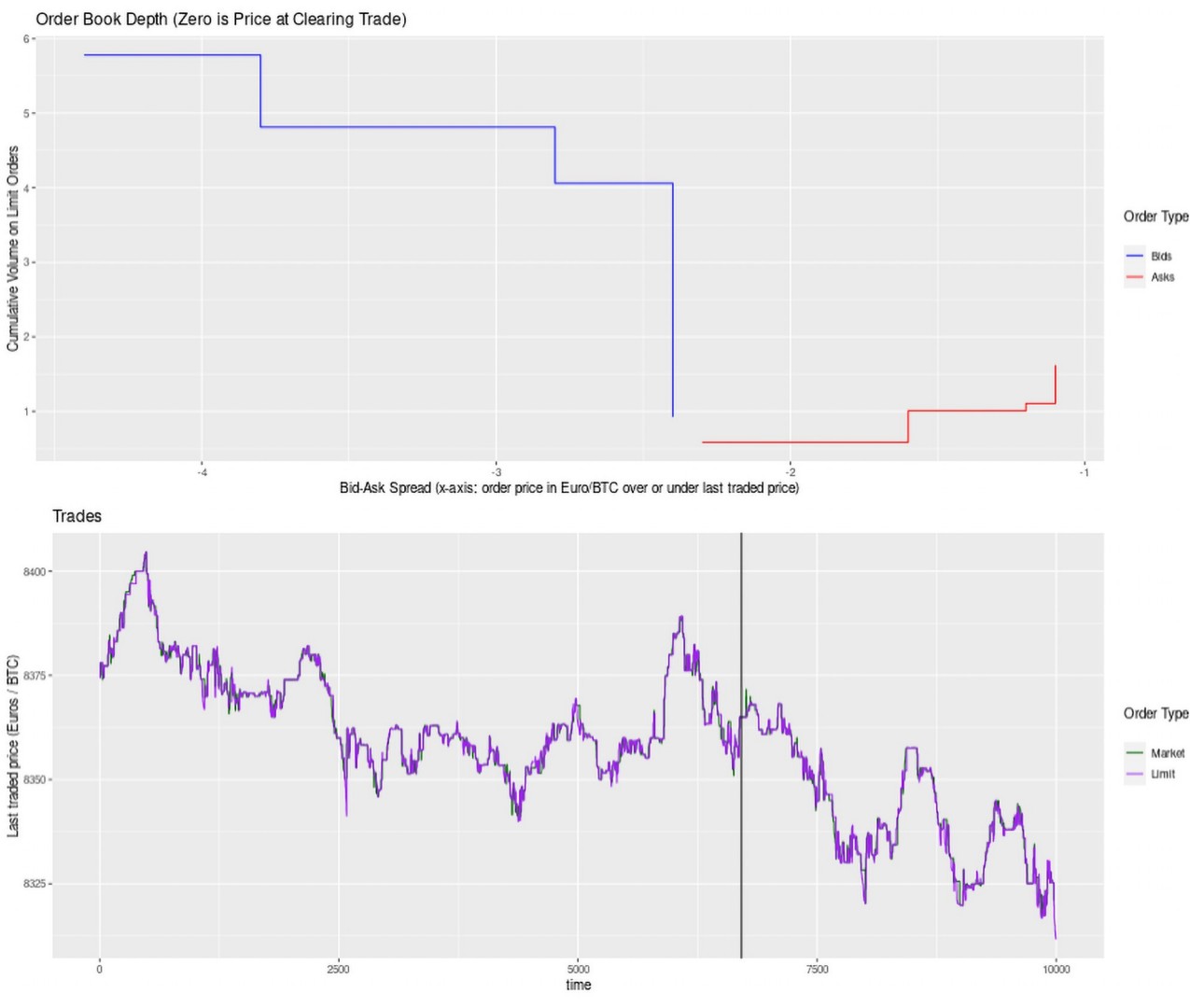

**Fig 6. Best four orders on either side and market price.**

Generalized method of moments [17] estimation was applied to an LOB model restricted to the four best quotes on both sides yielding 13 moment conditions: 4 break-even (5), 8 updating (6), and 1 market order size (7). Time ticks represent the arrival of a market order, and the LOB state is shown just ahead of the next market order arrival. The time between market orders is assumed to be sufficient for limit order posters to adjust their positions. This makes sense in the Kraken Bitcoin market where ~20 limit orders are submitted for every market order.

## 5. Empirical fit and tests

I ran the model against four samples of approximately 2 hours each, one overnight sample of 8 hours, and one baseline sample of 32 hours of Bitcoin data from the limit order book and market trade data in early August 2020 (Tables 2 and 3).

The baseline sample preceded the August 6 2020 date that I predicted to be "information rich" where a bitcoin 'whale' transferred 92857 BTC (≈$1.1 *billion*) between two wallets,

**Table 2. Parameter estimates for each run.**

| Run | alpha | (t-stat) | gamma | (t-stat) | c | (t-stat) | lambda | (t-stat) | J-stat |
|---|---|---|---|---|---|---|---|---|---|
| Base (32 hrs) | **0.598** | *113.747* | **32.777** | *99.787* | **-34.361** | *-97.056* | **58.386** | *150.802* | 624.64 |
| 1st (8 hrs) | **0.782** | *36.465* | **21.838** | *21.470* | **-25.246** | *-20.991* | **30.152** | *22.815* | 570.74 |
| 2nd (2 hrs) | **0.472** | *55.423* | **19.520** | *54.109* | **-20.995** | *-51.864* | **43.519** | *82.800* | 1358.00 |
| 3rd (2 hrs) | **0.639** | *46.315* | **26.193** | *34.772* | **-27.803** | *-30.356* | **43.480** | *57.881* | 1352.90 |
| 4th (2 hrs) | **0.405** | *32.971* | **21.084** | *32.911* | **-24.072** | *-47.459* | **55.513** | *50.800* | 1571.90 |
| 5th (2 hrs) | **0.325** | *58.923* | **17.048** | *53.537* | **-18.361** | *-45.280* | **55.763** | *73.537* | 1941.90 |

**Table 3. Dataset size and dates for each run.**

| Run | Total Orders | Total Trades | First Order Time | Last Order Time |
|---|---|---|---|---|
| Base (32 hrs) | 43974000 | 8943200 | 2020-08-02 21:11:50 MST | 2020-08-04 07:47:53 MST |
| 1st (8 hrs) | 9766000 | 2000000 | 2020-08-07 21:47:30 MST | 2020-08-08 00:05:37 MST |
| 2nd (2 hrs) | 9766000 | 2000000 | 2020-08-07 11:24:19 MST | 2020-08-07 13:42:47 MST |
| 3rd (2 hrs) | 9766000 | 2000000 | 2020-08-07 15:06:07 MST | 2020-08-07 17:22:06 MST |
| 4th (2 hrs) | 9766000 | 2000000 | 2020-08-07 18:43:42 MST | 2020-08-07 20:59:41 MST |
| 5th (2 hrs) | 9766000 | 2000000 | 2020-08-08 06:19:14 MST | 2020-08-08 08:46:09 MST |

setting off several days of speculation [15]. This is intended to give an idea of "normal" values of the estimated parameters.

All estimates t-statistics and *J*-statistics (*J* is distributed $\chi_9^2$) were significant at <.001 level.

## 6. Conclusions

The following table (Table 4) compares the estimators from this research to empirically fit the [5] model to data, to corresponding estimators from the two prior studies that attempted empirical fit in [14, 18].

To some extent, the three rows compare 'apples to oranges' as transaction sizes and prices are substantially different between a share of stock which is likely to have prices under $100 and volumes in the 10s and 100s; to Bitcoin, which has prices over $10,000 and volumes in fractions of one Bitcoin. The scale (price and transaction volume) is substantially different in the three different datasets, causing the large differences in the $\lambda$ and $c$ estimates. The $\alpha$ estimates should be consistent. The small $\gamma$ in the current research reflects the cost of trading Bitcoin.

Compared with the current study, both [14, 18] fit relatively small datasets, and estimated the transaction cost $\gamma$ to be significantly negative, and the fit statistics were poor. The small sample size is of special concern in GMM estimation, since GMM estimators tend to be strongly biased for small samples. Indeed, the results suggested this, as the *J-stats* generally rejected Glosten's model, and a key estimate of trading commission was negative. This is not surprising as the generalized method of moments requires a substantial volume of transactions

**Table 4. Glosten's structural model results compared.**

| | $\alpha$ | $\gamma$ | $c$ | $\lambda$ |
|---|---|---|---|---|
| Current Research averages | 0.53683 | 23.07667 | -25.13967 | 47.80217 |
| [18] | 0.01 | -0.01 | 1.38 | 0.03 |
| [14] | 2.60 | -0.99 | 11.17 | 11.45 |

to converge, and it is likely these prior studies performed poorly due to insufficient volume of data. To better understand the consequences of the current research findings, it is useful to recap what the estimated parameters mean in the context of the equilibrium conditions of the Glosten model. Recall that $c$ is simply the intercept in the equation computing the true value of the cryptocurrency $v_t = c + v_{t-1} + \alpha X_t + \eta_t$ in terms of the trade volume. It really could be any value without altering our interpretation of key empirical results. GMM's *J-test* statistic provides a measure to test the over-identifying restrictions where there are more moments than parameters (in this case 13 moments to estimate 4 parameters). The *J-test* is a Wald statistic under the null $H_0$: $E[g(\theta_0, x_i)] = 0$ and has a large sample $\chi^2$ distribution with $13 - 4 = 9$ degrees of freedom.

I argue that my data is consistent with Glosten's assumptions for two reasons. First, limit order trading and market-making is more attractive because cryptocurrency fees and commissions are rapidly approaching zero, and are orders of magnitude smaller than fees charged in the 1990s at the DAX and SSE. Second, limit order activity is rapidly increasing because of radically lower fees and technological development. [14] ratio of limit to market orders was 1.7; asset markets typically have ratios of 5 to 10; my Kraken-Bitcoin ratio of limit to market orders is around 20.

## 7. Discussion

[5] derived equilibrium prices of bids and asks in an electronic open limit order book, predicting that:

1. the order book would have a small-trade positive bid-ask spread, where limit orders profit from small traders

2. such an LOB exchange would provide as much liquidity as possible in extreme situations,

3. the LOB would discourage competition from third market dealers, and

4. if a trade earns positive profits, the prices will match those in the limit order book price schedule.

Glosten's model was developed in the early 1990s, at a time of rapid innovation in electronic markets. Data sources, markets and methods were insufficiently developed to provide reliable empirical tests of the model. By the end of the decade, though [14], was able to test [5] using limited data from the Stockholm Stock Exchange; ultimately rejecting a model yielding counter-intuitive estimators; e.g., the SSE data transaction costs were estimated to be negative. [18] fit Glosten's model to data from the thirty German DAX stocks using generalized method of moments (GMM) estimators, rejecting Glosten's model in 29 out of the 30 German DAX stocks.

The large volume of Bitcoin trades analyzed in the current research has allowed fitting the relatively complex [5] model with GMM, where prior models had generated biased and counter-intuitive estimates. In comparison the SSE and DAX are relatively small markets without a wide base of traders, and where insider trading dampens liquidity and discourages wide ownership of assets. The DAX stocks average only 2000 trades a day and SSE experienced even lower volumes. [5] defines $\alpha$ to depict the information contained in a market order in terms of its impact on the LOB. The current research strongly supports the [5] modeling assumptions; furthermore, even though the order book depth is only approximated with the four best orders on either side, $\lambda$ estimates that the model overall is capturing much of the information in the LOB.

The comparatively large size of λ in the Bitcoin estimates reflects the fact that GMM estimation was based on the prices and volumes of the four best orders on either side. But the Kraken Bitcoin book typically has a depth of 50-100 orders on either side, and though four best orders may be enough to characterize the market, the expected volumes, i.e., $E(|X_t|)$, will be much larger than just that contained in the eight best orders. A GMM estimation model, though, to estimate the full book would need 100 to 200 moment conditions, which consumes many degrees of freedom while adding little information to the estimation. Thus I argue that the 'four best orders on either side' formulation provides a suitable trade-off between estimation and full-information.

The current research overcame limitations of these two prior tests of Glosten's model by massively increasing the size of the dataset, and concentrating on one significant, highly liquid cryptocurrency asset: Bitcoin, which accounts for over half of the cryptocurrency markets capitalization. [18] observed that SSE market structure is substantially at variance with Glosten's assumptions The DAX 30, in turn, are traded markets subject to Germany's very lax enforcement of insider trading; traditionally in both German Corporation and Stock Exchange Law there have been no provisions against insider trading. German stocks may also have large ownership shares controlled by labor unions and wealthy families, which further distorts trading behavior.

Parameter $\alpha$ is a scaling parameter that reflects the relative effect of new information on price changes versus LOB volumes. On the updating restriction condition, these are price changes since the last market trade; on the break-even conditions they are the LOB bid-ask spread of the $k^{th}$ best order on either side of the market. A larger $\alpha$ means that new information results in larger price changes at a given volume. $\alpha$ is the key modeling parameter for Bitcoin market order informativeness. If $\alpha$ is large, then the price change, and the steepness of the depth chart for the LOB will be large—the market order will have injected a large amount of information into the market, which will create a large move in Bitcoin price which is reflected in both realized price, and in LOB prices.

The time frames in which each of these datasets were gathered is important in interpreting the value of $\alpha$. $\alpha$ is largest (7.82) in the $1^{st}$ run, which was an overnight 8 hours of data from the evening of August 7-8. This is an information rich time period, as it reflects international trades on the U.S. based Kraken platform, and may also reflect overnight portfolio adjustments by algorithmic traders. To a lesser extent, the afternoon August 8, 3pm to 5pm trades display a high $\alpha$ (0.639) potentially reflecting end-of-day portfolio adjustments. Portfolio adjustments reflect relative prices and returns on all assets, and some such as bonds and equities may be limited to certain trading hours. Fig 7's snapshot of Kraken's history shows that Bitcoin prices, which had just crossed the $11,000 mark, were quite volatile during that period, thus there was substantial information (and misinformation) to impart to the market. These observations are suggestive, but require further study before relying on them.

The $\gamma$ parameter reflects an equilibrium value of the platform services that facilitate a trade. Kraken's fee schedules are volume-based and are calculated as a percentage of the trade's quote currency volume ranging from 0% to.26% of the value of the trade. Our empirical results gave a value of $\gamma$ that averaged $23 per trade. In early August 2020, average trade price was around $11,000 and average trade volume was around 55 Bitcoin, suggesting that this value of $\gamma$ is commensurate with a commission rate of $\approx \frac{23}{11000 \times 55} \approx 0.000038\%$. This "equilibrium" rate was computed on averages; a more accurate figure would need to consider the full distribution of trade sizes and commission rates.

Prior studies found the transaction cost parameter $\gamma$ to be significantly negative (i.e., the platform was actually paying traders to make trades, rather than charging them commissions).

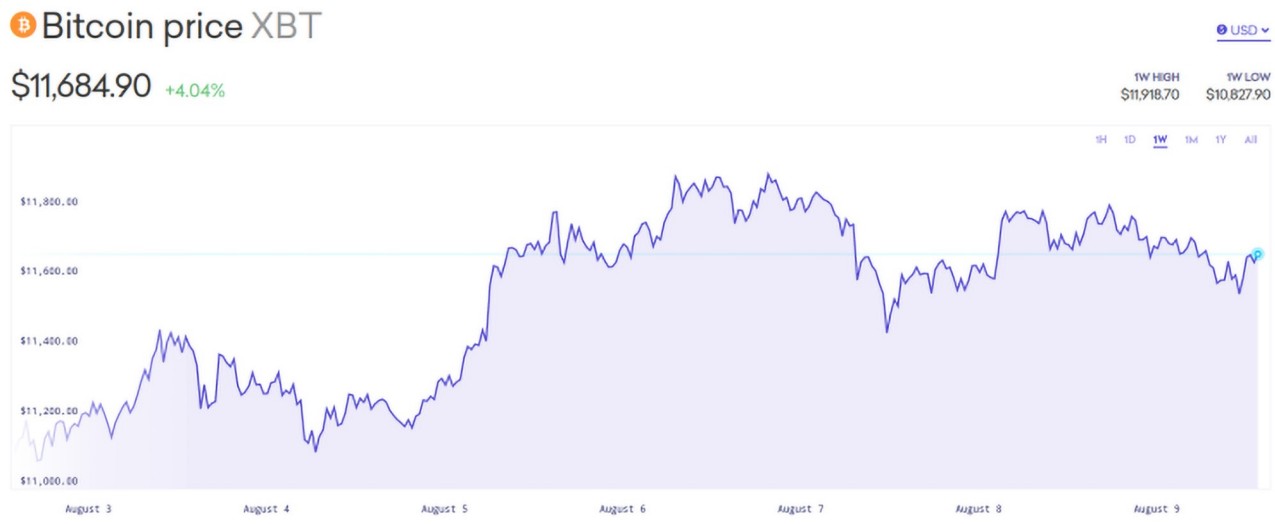

**Fig 7. Bitcoin/$ prices during August 7 to 9.**

The current research, in contrast, is the first market analysis of Glosten's structural model to yield a positive, and credibly large transaction cost parameter $\gamma$. The actual value of $\gamma$ can be difficult to discern. Kraken's fee schedules are volume-based and are calculated as a percentage of the trade's quote currency volume ranging from 0% to.26% of the value of the trade, and the average of $23 if consistent with Kraken's fee schedule.

[18] produced some very strange values of $\alpha$ and $\gamma$ in their study, leading one to question whether their dataset actually represented an efficient market. Their research implies that for a vigorously traded cryptocurrency such as Bitcoin, market orders should be relatively uninformative. The average value of $\alpha$ in this study of 0.54 is smaller than [14] but larger than [18].

The $\lambda$ parameter is simply the expected trade size from the market order size condition. The empirical value seems approximately two magnitudes larger than the directly computed $E(|X_t|) \approx 0.20$ leading us to question what is happening. It is important to remember that trade volumes are indigenous to the trading platform, but prices are set across all platforms in the market, and any differences would quickly be arbitraged away. It is also likely that very large trades may be facilitated outside of any platform by a direct wallet-to-wallet transfer. The empirical value of $\lambda \approx 55$ might indeed reflect an empirical value commensurate with the price —i.e., off platform, wallet-to-wallet trades may be driving the prices in the market. Perhaps more conspiratorially, large traders ("whales") may actually be manipulating price changes in direct wallet-to-wallet transfer, and these later trickle-down to on-platform trading through a large number of profit-taking small market orders.

I interpret these results as strong support that [5]'s seminal theoretical models of electronic limit order book markets are valid models to explain liquidity, equilibrium and information asymmetries in Bitcoin markets. The current study did not specifically look at the cross-section; future studies will need to compare Bitcoin to other cryptocurrencies, ideally controlled on a single platform such as Kraken.

## Author Contributions

**Conceptualization:** J. Christopher Westland.

**Data curation:** J. Christopher Westland.

**Formal analysis:** J. Christopher Westland.

**Investigation:** J. Christopher Westland.

**Methodology:** J. Christopher Westland.

**Project administration:** J. Christopher Westland.

**Resources:** J. Christopher Westland.

**Software:** J. Christopher Westland.

**Supervision:** J. Christopher Westland.

**Validation:** J. Christopher Westland.

**Visualization:** J. Christopher Westland.

**Writing – original draft:** J. Christopher Westland.

**Writing – review & editing:** J. Christopher Westland.

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
