## [Decision Letter · Decision Letter 0]

9 Apr 2021

PONE-D-21-06529

Trade Informativeness and Liquidity in Bitcoin Markets

PLOS ONE

Dear Dr. Westland,

Thank you for submitting your manuscript to PLOS ONE. After careful consideration, we feel that it has merit but does not fully meet PLOS ONE’s publication criteria as it currently stands. Therefore, we invite you to submit a revised version of the manuscript that addresses the points raised during the review process.

At this moment, two referees have revised your manuscript and both of them agree that your research is enough interesting and novel to deserve publication in PLOS ONE. So do I. Also, both reviewers have made a short list of minor corrections and suggestions to include in a revised version of your manuscript. These corrections rank  from correcting some typos to suggestions in the way you are presenting your results. 

We look forward to receiving your revised manuscript.

Kind regards,

Alejandro Raul Hernandez Montoya, Ph D

Academic Editor

PLOS ONE

Journal Requirements:

1. Please ensure that your manuscript meets PLOS ONE's style requirements, including those for file naming. The PLOS ONE style templates can be found athttps://journals.plos.org/plosone/s/file?id=wjVg/PLOSOne_formatting_sample_main_body.pdf and https://journals.plos.org/plosone/s/file?id=ba62/PLOSOne_formatting_sample_title_authors_affiliations.pdf

2. Thank you for submitting the above manuscript to PLOS ONE. During our internal evaluation of the manuscript, we found significant text overlap between your submission and the following previously published works, some of which you are an author.

http://publikationen.ub.uni-frankfurt.de/files/21084/11_09.pdf

https://decrypt.co/37171/lost-bitcoin-3-7-million-bitcoin-are-probably-gone-forever

Please revise the manuscript to rephrase the duplicated text, cite your sources, and provide details as to how the current manuscript advances on previous work. Please note that further consideration is dependent on the submission of a manuscript that addresses these concerns about the overlap in text with published work.

4. Please ensure that you refer to Figures 3 to 6 in your text as, if accepted, production will need this reference to link the reader to the figure.

Additional Editor Comments (if provided):

Reviewers' comments:

Reviewer's Responses to Questions

**Comments to the Author**

1. Is the manuscript technically sound, and do the data support the conclusions?

Reviewer #1: Yes

Reviewer #2: Yes

2. Has the statistical analysis been performed appropriately and rigorously? 

Reviewer #1: Yes

Reviewer #2: Yes

3. Have the authors made all data underlying the findings in their manuscript fully available?

Reviewer #1: Yes

Reviewer #2: Yes

4. Is the manuscript presented in an intelligible fashion and written in standard English?

Reviewer #1: Yes

Reviewer #2: Yes

5. Review Comments to the Author

Reviewer #1: The paper is well written and interesting to read. In my opinion is suitable of publication is Plos one but I have some minor comments. The structure must re-done. Introduction section is too big and conclusion section also. I suggest to the authors to split this section into two sections: conclusions and discussions.

Conclusion section must only summarize the findings of the manuscript against to the current literature on this topic.

Reviewer #2: The author studies the Glosten's Structural Model about liquidity and the limit order book in Bitcoin markets. I think the paper is well structured and the results are interesting, so I recommend it for publication. However, minor misprints should be revised, for example:

- pg. 3: but know one can know

- pg. 16: and and

- pg. 19: market A larger

- pg. 19: the the

6. PLOS authors have the option to publish the peer review history of their article (what does this mean?). If published, this will include your full peer review and any attached files.

Reviewer #1: No

Reviewer #2: No

---

## [Author Response · Author response to Decision Letter 0]

24 May 2021

## Comments from the AE

Thank you for submitting your manuscript to PLOS ONE. After careful consideration, we feel that it has merit but does not fully meet PLOS ONE’s publication criteria as it currently stands. Therefore, we invite you to submit a revised version of the manuscript that addresses the points raised during the review process.

_Thank you for your kind words, and for a quick and insightful review of my submission. I have addressed all of the points raised by the editor and reviewers, as is documented in the responses below. _

A marked-up copy of your manuscript that highlights changes made to the original version. You should upload this as a separate file labeled 'Revised Manuscript with Track Changes'. An unmarked version of your revised paper without tracked changes. You should upload this as a separate file labeled 'Manuscript'.

_My revised 'Manuscript' in PDF format and my 'Revised Manuscript with Track Changes' in DOCX format have been uploaded for this revision. I have no changes to make in my financial disclosure. I use RMarkdown to write and format my documents, and unfortunately, revision edits have been a longstanding issue in both the rmarkdown and pandoc user group. Currently the best available approach (the one that I have used here) is to 'knit' the original and revised manuscripts to .DOCX files, and use LibreOffice Writer to generate the original with all of the markup annotations. This provides accurate records of all markups, but the document will have MSWord style formatting rather than the more aesthetically pleasing LaTeX formatting of the submission. I hope this is sufficient for the editor and reviewers on submitting my revision._

We recommend that you deposit your laboratory protocols in protocols.io to enhance the reproducibility of your results. Protocols.io assigns your protocol its own identifier (DOI) so that it can be cited independently in the future. 

_This was not a laboratory study, rather it was a 'natural experiment' where the data was extracted from the external Kraken trading platform database. In the spirit of encouraging replicability of my results, I have uploaded to a Kaggle repository, all of my data and code at https://www.kaggle.com/westland/bitcoin-mkt-liquidity-informativeness . The relevant DOI is James Christopher Westland, “Bitcoin Market Trade Informativeness and Liquidity.” Kaggle, 2021, doi: 10.34740/KAGGLE/DSV/2190153 _

Journal Requirements: When submitting your revision, we need you to address these additional requirements. Please ensure that your manuscript meets PLOS ONE's style requirements, including those for file naming. The PLOS ONE style templates can be found at https://journals.plos.org/plosone/s/file?id=wjVg/PLOSOne_formatting_sample_main_body.pdf and https://journals.plos.org/plosone/s/file?id=ba62/PLOSOne_formatting_sample_title_authors_affiliations.pdf

_In the current revision, I have used the https://github.com/rstudio/rticles template `plos.csl` for PLOS journals using RMarkdown under RStudio to reformat the revisions according to the PLOSOne_formatting template. The following YAML header, with additional HTML for Pandoc use was used to format to PLOSOne specifications:_

```

---

output:

 pdf_document:

 latex_engine: xelatex

 keep_tex: true

 word_document: default

 html_document:

 theme: null

always_allow_html: yes

csl: ~/Desktop/.../plos.csl

header-includes:

 \\usepackage{setspace}\\doublespacing

 \\usepackage{longtable}

 \\usepackage{float}

 \\usepackage{amsmath}

bibliography: ~/Desktop/.../order_inform.bibtex

abstract: "Liquid markets are driven by information asymmetries and the injection of new information in trades into market prices. Where market matching uses an electronic limit order book (LOB), limit orders traders may make suboptimal price and trade decisions based on new but incomplete information arriving with market orders. This paper measures the information asymmetries in Bitcoin trading limit order books on the Kraken platform, and compares these to prior studies on equities LOB markets. In limit order book markets, traders have the option of waiting to supply liquidity through limit orders, or immediately demanding liquidity through market orders or aggressively priced limit orders. In my multivariate analysis I control for volatility, trading volume, trading intensity and order imbalance to isolate the effect of trade informativeness on book liquidity. The current research, offers the first empirical study of Glosten (1994) to yield a positive, and credibly large transaction cost parameter. Trade and LOB datasets that were several orders of magnitude larger than any of the prior studies. Given the poor small sample properties of GMM, it is likely that this substantial increase in size of datasets is essential for validating the model. _J-stats_ and all other fit measures were significantl. The research strongly supports Glosten's seminal theoretical model of limit order book markets, showing that these are valid models of Bitcoin markets. This research empirically tested and confirmed trade informativeness as a prime driver of market liquidity in the Bitcoin market."

---

<style type="text/css">

body{ /* Normal */

 font-size: 12px;

 }

td { /* Table */

 font-size: 8px;

}

h1.title {

_Thank you for your kind words, and for a quick and insightful review of my submission. I have addressed all of the points raised by the editor and reviewers, as is documented in the responses below. _

 font-size: 38px;

 color: Black;

}

h1 { /* Header 1 */

 font-size: 18px;

 color: Black;

}

h2 { /* Header 2 */

 font-size: 16px;

 color: Black;

}

h3 { /* Header 3 */

 font-size: 14px;

 font-family: "Times New Roman", Times, serif;

 color: Black;

}

code.r{ /* Code block */

 font-size: 12px;

}

pre { /* Code block - determines code spacing between lines */

 font-size: 14px;

}

</style>

```

2. Thank you for submitting the above manuscript to PLOS ONE. During our internal evaluation of the manuscript, we found significant text overlap between your submission and the following previously published works, some of which you are an author.

http://publikationen.ub.uni-frankfurt.de/files/21084/11_09.pdf

https://decrypt.co/37171/lost-bitcoin-3-7-million-bitcoin-are-probably-gone-forever

_I apologize (and am a bit embarrassed) about the degree of overlap in descriptive and supporting text. I have aggressively re-edited overlapping portions of the paper. None of the overlapping sections pertained to the central empirical contributions of this paper. Instead they arose in part from my rather obsessive concern about firmly linking this paper to prior literature in the field. The http://publikationen.ub.uni-frankfurt.de/files/21084/11_09.pdf overlap wasn't actually from that working paper, but from the final published 'Beltran-Lopez H, Grammig J, Menkveld AJ. Limit order books and trade informativeness. The European Journal of Finance. 2012;18: 737–759' paper, which is repeatedly cited in my text, and was the source for the model that I empirically tested. From my past experience, finance reviewers tend to be particular about grounding research in prior models that have been tested (i.e., being part of the ongoing research conversation). I was relatively obsessive about anchoring my language in precisely that of the prior literature, and used their descriptions, always with attribution. Some other parts of the overlap comes from the R language descriptions of the algorithms that I have used to analyze the data, and again, accuracy of description was my intent. I have aggressively edited these portions of the text that overlap with prior work to paraphrase and shorten these descriptions, while maintaining the accuracy of my assertions. Additional overlaps arose from my attempts to characterize the structural nature of the Bitcoin market, using news sources, and my application of the generalized method of moments (where I relied on open source documents at https://cran.r-project.org/). I have removed sections, editing down the size of the introduction and final sections, as well as completely rewritten sections where this overlap was a problem. My current `iThenticate` review of the resubmitted manuscript shows 12% overlap with existing sources, most of these being generic descriptive phrases that appear in many publications._

_I have uploaded to a Kaggle repository, all of my data and code at https://www.kaggle.com/westland/bitcoin-mkt-liquidity-informativeness . The relevant DOI is James Christopher Westland, “Bitcoin Market Trade Informativeness and Liquidity.” Kaggle, 2021, doi: 10.34740/KAGGLE/DSV/2190153 _

4. Please ensure that you refer to Figures 3 to 6 in your text as, if accepted, production will need this reference to link the reader to the figure.

_The references to figures 3 to 6 have been made in the text_

_The references in the revision are complete and accurate. There have been no retractions of these referenced papers._

## Comments from Reviewers 1 and 2

Comments to the Author

1. Is the manuscript technically sound, and do the data support the conclusions? The manuscript must describe a technically sound piece of scientific research with data that supports the conclusions. Experiments must have been conducted rigorously, with appropriate controls, replication, and sample sizes. The conclusions must be drawn appropriately based on the data presented.

- Reviewer #1: Yes

- Reviewer #2: Yes

_Thank you for your kind words, and for a quick and insightful review of my submission. I have addressed all of the points raised by the editor and reviewers, as is documented in the responses below. _

2. Has the statistical analysis been performed appropriately and rigorously?

- Reviewer #1: Yes

- Reviewer #2: Yes

_Thank you and I appreciate your thorough review of my statistical modeling. _

3. Have the authors made all data underlying the findings in their manuscript fully available? The PLOS Data policy requires authors to make all data underlying the findings described in their manuscript fully available without restriction, with rare exception (please refer to the Data Availability Statement in the manuscript PDF file). The data should be provided as part of the manuscript or its supporting information, or deposited to a public repository. For example, in addition to summary statistics, the data points behind means, medians and variance measures should be available. If there are restrictions on publicly sharing data—e.g. participant privacy or use of data from a third party—those must be specified.

- Reviewer #1: Yes

- Reviewer #2: Yes

_I have uploaded to a Kaggle repository, all of my data and code at https://www.kaggle.com/westland/bitcoin-mkt-liquidity-informativeness . The relevant DOI is James Christopher Westland, “Bitcoin Market Trade Informativeness and Liquidity.” Kaggle, 2021, doi: 10.34740/KAGGLE/DSV/2190153 _ 

4. Is the manuscript presented in an intelligible fashion and written in standard English? PLOS ONE does not copyedit accepted manuscripts, so the language in submitted articles must be clear, correct, and unambiguous. Any typographical or grammatical errors should be corrected at revision, so please note any specific errors here.

- Reviewer #1: Yes

- Reviewer #2: Yes

_Thank you. In the current revision, I have spent time copy-editing both for typos and readability. I hope you will find the revised manuscript even better organized and written. _

# Review Comments to the Author

Reviewer #1: The paper is well written and interesting to read. In my opinion is suitable of publication is Plos one but I have some minor comments. The structure must re-done. Introduction section is too big and conclusion section also. I suggest to the authors to split this section into two sections: conclusions and discussions. Conclusion section must only summarize the findings of the manuscript against to the current literature on this topic.

_Thank you for your suggestions. Sometimes as an author you spend so much time with the material, that you miss final edits that would benefit the reader. I have done the following in response to your suggestions:_

_1. I have made a significant number of revisions to shorten the introductory material, while trying to maintain all of the prior studies and construct descriptions needed to support the delineation of the main model in the paper. _

_1. I have split the first revisions "Conclusion" into two parts: _

_a. a "Conclusion" section that only summarizes the findings of the manuscript with current literature on this topic._

_b. a "Discussion" section that reviews implications of the conclusions, and suggests additional research that can provide a deeper understanding of cryptocurrency markets in the future._

_I believe these revisions provide a significant improvement in communication and readability over the first draft, and I hope the reviewer agrees._

Reviewer #2: The author studies the Glosten's Structural Model about liquidity and the limit order book in Bitcoin markets. I think the paper is well structured and the results are interesting, so I recommend it for publication. However, minor misprints should be revised, for example:

- pg. 3: but know one can know

- pg. 16: and and

- pg. 19: market A larger

- pg. 19: the the

_These have been corrected. Additionally, I uploaded the paper to Grammarly for checking and found quite a few other typos. I have corrected all of these now. _

6. PLOS authors have the option to publish the peer review history of their article (what does this mean?). If published, this will include your full peer review and any attached files. Do you want your identity to be public for this peer review? For information about this choice, including consent withdrawal, please see our Privacy Policy.

- Reviewer #1: No

- Reviewer #2: No

_Thank you for a quick and insightful review of my submission. _

_I have uploaded the revision to https://pacev2.apexcovantage.com/Upload where PACE found no problems with my revision_

---

## [Decision Letter · Decision Letter 1]

19 Jul 2021

Trade Informativeness and Liquidity in Bitcoin Markets

PONE-D-21-06529R1

Dear Dr. Westland,

We’re pleased to inform you that your manuscript has been judged scientifically suitable for publication and will be formally accepted for publication once it meets all outstanding technical requirements.

Kind regards,

Alejandro Raul Hernandez Montoya, Ph D

Academic Editor

PLOS ONE

Additional Editor Comments (optional):

Reviewers' comments:

Reviewer's Responses to Questions

**Comments to the Author**

1. If the authors have adequately addressed your comments raised in a previous round of review and you feel that this manuscript is now acceptable for publication, you may indicate that here to bypass the “Comments to the Author” section, enter your conflict of interest statement in the “Confidential to Editor” section, and submit your "Accept" recommendation.

Reviewer #1: All comments have been addressed

Reviewer #2: All comments have been addressed

2. Is the manuscript technically sound, and do the data support the conclusions?

Reviewer #1: Yes

Reviewer #2: Yes

3. Has the statistical analysis been performed appropriately and rigorously? 

Reviewer #1: Yes

Reviewer #2: Yes

4. Have the authors made all data underlying the findings in their manuscript fully available?

Reviewer #1: Yes

Reviewer #2: Yes

5. Is the manuscript presented in an intelligible fashion and written in standard English?

Reviewer #1: Yes

Reviewer #2: Yes

6. Review Comments to the Author

Reviewer #1: (No Response)

Reviewer #2: I recommend the paper for publication. In the new version there are a few new typos, like "Cryoptocurrency" or fishes" instead of "fishes".

On the other hand, the notation ⟂⟂ does not seem standard to me, I would better revert it to is independent of .

7. PLOS authors have the option to publish the peer review history of their article (what does this mean?). If published, this will include your full peer review and any attached files.

Reviewer #1: No

Reviewer #2: No

---

## [Editor Report · Acceptance letter]

2 Aug 2021

PONE-D-21-06529R1 

Trade Informativeness and Liquidity in Bitcoin Markets 

Dear Dr. Westland:

I'm pleased to inform you that your manuscript has been deemed suitable for publication in PLOS ONE. Congratulations! Your manuscript is now with our production department. 

Kind regards, 

on behalf of

Dr. Alejandro Raul Hernandez Montoya 

Academic Editor

PLOS ONE